# The Dangerous Liaisons in the Oxidative Stress Response to *Leishmania* Infection

**DOI:** 10.3390/pathogens11040409

**Published:** 2022-03-28

**Authors:** Marta Reverte, Tiia Snäkä, Nicolas Fasel

**Affiliations:** Department of Biochemistry, Faculty of Biology and Medicine, University of Lausanne, 1066 Lausanne, Switzerland; marta.reverteroyo@unil.ch (M.R.); tiia.snaka@unil.ch (T.S.)

**Keywords:** macrophage, ROS, inflammation, NRF2, NF-kB, *Leishmania*, LRV1

## Abstract

*Leishmania* parasites preferentially invade macrophages, the professional phagocytic cells, at the site of infection. Macrophages play conflicting roles in *Leishmania* infection either by the destruction of internalized parasites or by providing a safe shelter for parasite replication. In response to invading pathogens, however, macrophages induce an oxidative burst as a mechanism of defense to promote pathogen removal and contribute to signaling pathways involving inflammation and the immune response. Thus, oxidative stress plays a dual role in infection whereby free radicals protect against invading pathogens but can also cause inflammation resulting in tissue damage. The induced oxidative stress in parasitic infections triggers the activation in the host of the antioxidant response to counteract the damaging oxidative burst. Consequently, macrophages are crucial for disease progression or control. The ultimate outcome depends on dangerous liaisons between the infecting *Leishmania* spp. and the type and strength of the host immune response.

## 1. Introduction

During the blood meal of an infected sand fly, *Leishmania* (*L.*) parasites are injected into the mammalian host and are internalized by phagocytic cells, principally macrophages. In macrophages, *Leishmania* promastigotes differentiate into obligate intracellular amastigotes [1]. Amastigotes reside in phagolysosome-like organelles, where they survive and replicate [2]. The phagolysosomes present an increase in temperature and an acidic environment, which trigger the differentiation of promastigotes to amastigotes [3]. They can either host numerous or single amastigotes, depending on the infecting species [4]. Amastigotes replicate until the rupture of the macrophage, the released amastigotes are then internalized by the surrounding phagocytes, leading to the expansion of the infection.

*Leishmania* spp. principally affect the skin and/or the mucosal tissues depending on the infecting species, which are determinants for the type of cutaneous outcome and clinical pathology, but also for the host inflammatory and anti-inflammatory response [5]. Localized cutaneous leishmaniasis (LCL) remains the most prevalent clinical manifestation of leishmaniasis. It is normally non-life-threatening but can be often associated with a social stigma.

In murine experimental models, protection against cutaneous leishmaniasis is associated with a robust T helper (Th) 1 (Th1) cell immune response and the production of interferon (IFN) gamma (IFN-γ) and tumor necrosis factor-alpha (TNF-α), whereas susceptibility is associated with a Th2 response involving IL-10 and IL-4. In humans, the situation is not as clear and there is a large spectrum of immunological responses with different levels in T cell responses and IFN-γ [5,6]. In any case, IFN-γ is central in the defense against *Leishmania* and boosts parasite control by activating infected macrophages to induce microbicidal effectors to enhance parasite killing [7]. In LCL, the early immune response is mediated by T-cell derived TNF-α and IFN-γ, however, at later stages, Th2 cells producing interleukin (IL)-10 and transforming growth factor-beta (TGF-β) are detected, which could be attributed to a decrease in the proinflammatory cytokine storm once the disease is resolved [5].

In Latin America, infection with *L. guyanensis* or *L. braziliensis* mainly leads to self-healing LCL, however, between 5–10% of these infections may disseminate and manifest as metastatic forms such as mucocutaneous leishmaniasis (MCL) [5,8]. The MCL outcome is distinguished by its persistent, dormant, and metastatic behavior, in which parasites disseminate and secondary distant lesions appear, mostly in the oral and nasopharyngeal parts of the face. These clinical features of MCL can be highly disfiguring since parasite dissemination is followed by extensive tissue destruction linked to high immune cell infiltration leading to hyperinflammation [9,10]. Contrary to LCL, MCL lesions are not self-healing and require drug treatment [5,11]. Patients affected by MCL present high levels of TNF-α but decreased levels of IL-10 in comparison to LCL, which results in the hyperinflammatory response characteristic of MCL lesions, whereas the actual detectable parasite load is low [12]. Additionally, the IL-17 inflammation-inducing cytokine and Th17 cells are highly expressed in MCL lesions in comparison to LCL lesions [13].

Another metastatic form developed after LCL is DL (also called disseminated cutaneous leishmaniasis, DCL) and can be caused by *L. braziliensis*, *L. panamensis*, *L. guyanensis*, or *L. amazonensis* parasites [5]. Parasite dissemination in DL occurs within weeks or days after the initial lesion formation [14,15]. The characteristic clinical picture of DL consists of the formation of multiple nodules, papules, and ulcerated lesions starting at the infection site, which subsequently disseminates preferentially to the limbs. Additionally, MCL nasal mucosal lesions are also found in DL cases [16]. Due to the high number of lesions, DL is difficult to treat. A high pro-inflammatory Th1 response at the lesion site is detected but not in the peripheral blood in DL patients, which suggests that the decreased peripheral Th1 response could allow the spread of the parasite [16].

The presence of a viral endosymbiont, the *Leishmania* RNA virus (LRV) in the cytoplasm of some *Leishmania* species has been described already some years ago and may be considered a risk factor for the progression towards exacerbated forms of the disease including MCL and DL [17,18]. LRV belongs to the *Totiviridae* family, whose members are characterized by icosahedral particles present in a wide range of protozoa including *Trichomonas vaginalis*, *Entamoeba,* and *Toxoplasma gondi* [19,20,21,22]. The viral particles range between 30–40 nm in diameter composed of a non-segmented double-stranded RNA genome encoding a capsid protein and a capsid-RNA-dependent RNA polymerase fusion protein, crucial for the dsRNA virus replication [23]. LRV was first described in *L. Viannia* subgenus in the *L. guyanensis* strain [24] and subsequently in the *L. braziliensis* strain [25]. Additionally, it has also been detected in *L*. *Leishmania* subgenus in *L. major* [26], *L. infantum* [27], and *L. aethiopica* [28] strains. The LRV sequence varies between the two *L.* subgenera, therefore they have been differently categorized as LRV1 and LRV2 in *L. Viannia* and *L. Leishmania*, respectively. For example, *L. guyanensis* parasites are called *Lgy*LRV1+ or *Lgy*LRV1- depending on the presence of the LRV1 particles [29]. LRV1 presence in human *L. guyanensis* and *L. braziliensis* infection has been significantly associated with treatment failure and relapse. However, the mechanisms by which LRV1 modulates treatment failure have not as yet been described [30,31]. Taken together, both host and parasite factors will determine the outcome of the disease.

## 2. The Oxidative Stress Response

Oxidative stress was first reported 30 years ago and describes the imbalance between oxidants and antioxidants in favor of the oxidants, which results in failure of the redox signaling and consequent cell damage [32]. Oxidative stress arises from the excessive release of reactive oxygen species (ROS) and reactive nitrogen species (RNS) [33]. ROS avidly interacts with a broad variety of molecules such as proteins, nucleic acids, lipids, and carbohydrates. Through such interactions, ROS contributes to damage in the biological systems. ROS, in addition, are a key cellular defense against invading pathogens [34]. In response to intracellular pathogens such as *Leishmania* parasites, macrophages rapidly induce an oxidative stress response as a mechanism of defense to induce pathogen clearance and activate signaling pathways associated with inflammation and immune responses [35,36].

The superoxide, O_2_^−^, is the most common oxygen free radical. It can be produced as a byproduct in the mitochondria (Figure 1). The formation of O_2_^−^ derives from the electrons transfer along the different enzymes of the respiratory chain, which is not totally effective and leads to the leakage of electrons onto molecular oxygen resulting in O_2_^−^ [37,38]. Superoxide can also be produced from the leakage of electrons through the electron transport chain within the endoplasmic reticulum (ER) [39] or by 5-lipoxygenase [40]. Other oxygen radicals are the hydroxyl (•OH), the peroxyl (RO•2), and the alkoxyl (RO•). The non-radical intermediates that are either oxidizing agents and/or are simply transformed into radicals include hypochlorous acid (HOCl), ozone (O3), singlet oxygen (^1^O_2_), and hydrogen peroxide (H_2_O_2_). Superoxide is detoxified by the family of enzymes known as superoxide dismutase (SOD), responsible for its transformation into H_2_O_2_, which is less reactive than the free radical and which can also participate in signaling pathways [34]. SOD can be cytoplasmic (SOD1) or mitochondrial (SOD2). H_2_O_2_ produced by SODs is then detoxified into O_2_ and H_2_O by peroxiredoxins. ROS can also be generated by the phagocyte NOX enzymes, whose primary function is ROS production [37]. In the human genome, there are 7 NOX homologs: NOX1 to NOX5 and DUOX1 and DUOX2, which vary in their expression level, organ-specific expression, ROS release, and regulation of their activity [41].

NOX2 (also known as gp91*^phox^*) is the prototype of NADPH oxidases and thus the best-characterized isoform [42]. A complex series of protein/protein interactions are responsible for the activation of NOX2. In macrophages, NOX2 comprises the principal source of ROS. NOX2 is composed of six hetero-subunits, which connect in response to a stimulus to activate the enzyme complex and consequently produce superoxide (Figure 1) [43]. The two NOX2 subunits gp91*^phox^* and p22*^phox^* are integral membrane proteins that together comprise the large heterodimeric subunit called flavocytochrome b558 (cyt b558). The multidomain regulatory subunits, p40*^phox^*, p47*^phox^,* and p67*^phox^* exist in the cytosol as a complex under basal conditions. Upon stimulation, p47*^phox^* is phosphorylated and the whole complex is translocated to the membrane where it interacts with cyt b558 to compose the activate oxidase enzyme. To be active, the complex needs two low-molecular-weight guanine nucleotide-binding proteins, called Rac2 and Rap1A. Rac2 binds guanosine triphosphate (GTP) and translocates to the membrane along with the p40*^phox^*, p47*^phox^,* and p67*^phox^* complex. During the process of phagocytosis, the plasma membrane is internalized and becomes the interior membrane of the phagocytic vesicle leading to the release of O_2_•^−^ through the enzyme complex [34].

RNS comprises nitrogen-containing oxidants, such as NO• and its by-products including nitrate (NO_3_^−^), nitrite (NO_2_^−^), and peroxynitrite (ONOO•) [44]. In the organism, NO• is synthesized from L-arginine and molecular oxygen using NADPH as an electron donor. Overall, the reaction involves a two-step oxidative conversion of L-arginine to NO and L-citrulline via N-hydroxy-L-arginine as an intermediate (Figure 1) [45]. The enzymes responsible for NO• generation are nitric oxide synthases (NOSs). There are three different subtypes of NOS enzymes depending on the tissue type: eNOS (endothelial NOS), iNOS (inducible NOS), and nNOS (neuronal NOS). The iNOS isoform is the most relevant to phagocyte-pathogen interactions [46,47]. Stimulation of pattern recognition receptors (PRRs) together with signaling from pro-inflammatory cytokines can lead to iNOS transcription. In addition, the L-arginine substrate used by iNOS for generating NO• is also used by *Leishmania* parasites for the production of their essential nutrients L-ornithine and urea, resulting in the decrease of NO• and therefore favoring parasite survival [48]. Arginase 1 is the cytosolic enzyme responsible to convert L-arginine into urea and ornithine. Ornithine is a precursor of polyamines, which induce the synthesis of trypanothione and the proliferation of parasites. Trypanothione is an analog of glutathione essential for parasite protection against oxidants [49,50].

## 3. Oxidative Stress as Host Defense against *Leishmania*

In infection, the role of oxidative stress is dual: free radicals serve as protection against invading pathogens but consequently can also lead to inflammation resulting in tissue damage [51]. Phagocytic cells, such as macrophages and neutrophils, induce the antimicrobial response when infected by pathogens. The free radicals from oxygen, nitrogen, and chlorine derived from the macrophage respiratory burst are toxic to *Leishmania.* The first and foremost player among the prooxidants is the superoxide anion produced by the membrane-bound NOX2. This is the initial molecule of a storm of free radicals resulting in an oxidant milieu aiming at parasite killing. The use of mouse strains deficient in the gp91^phox^ (*gp91*^phox−/−^) subunit of NOX2 complex revealed that ROS are crucial for parasite killing. For instance, *gp91*^phox−/−^ mice infected with *L. amazonensis* presented severe pathology at the later stage of infection [52]

Nonetheless, *Leishmania* parasites have evolved strategies to antagonize the host immune system, therefore, contributing to their persistence and proliferation within macrophages [35]. Lipophosphoglycans (LPGs) molecules on the *Leishmania* surface have been described to inhibit the phosphorylation of the p47*^phox^* subunit and therefore block superoxide generation by NOX2 [53]. The induced oxidative stress in parasitic infections triggers the activation in the host of the antioxidant response to counteract the damaging oxidative burst. Such a response is principally mounted by the nuclear factor-erythroid 2-related factor 2 (NRF2) transcription factor leading to the decrease of oxidative free radicals, that consequently favor parasite persistence. 

## 4. The Antioxidant Stress Response: The NRF2 Transcription Factor

Infection induces oxidative stress and inflammation, whereas the host immune system induces the antioxidant and anti-inflammatory response as a mechanism of defense to limit infection and favor pathogen clearance. In this regard, the cytoprotective role of NRF2 has been widely studied as a therapeutic strategy for protection against viruses such as influenza virus [54] or *Leishmania* parasites [55].

NRF2 was first described in the early 90s as a protein that recognizes the NF-E2 binding site of human β-globin genes (Figure 2) [56]. Some years later the role of NRF2 was described in the antioxidant response evidenced by the transcriptional modulation of the *Nqo1* gene [57]. However, the detailed function of NRF2 in the antioxidant response was reported later by Itoh et al. [58]. This opened the door to thousands of studies involving NRF2 in the past decades. Indeed, from birth, animals must fight against multiple stressors that interfere with their homeostasis. Consequently, animals have been forced to evolve detoxifying systems like the NRF2 system that protects against a broad spectrum of stressors, including oxidative stress [59].

NRF2 is part of the cap’n’collar family of transcription factors, a family of basic leucine zipper transcription factors broadly conserved from worms to humans, but not present in plants or fungi [60]. Currently, NRF2 is related to a wide range of diseases in the field of inflammation, cancer, and metabolism [61,62,63]. Accordingly, NRF2 deficient (*Nrf2*^−/−^) mice have been described to be more vulnerable to chemical and radiation-induced tumorigenesis [62], exhibit more severe lung inflammation and damage upon exposure to cigarette smoke [64], and hyperoxia [65] in comparison to wild-type (WT) mice. The NRF2 stress response pathway is defined as the principal inducible defense against oxidative and electrophilic stresses. The antioxidant response controlled by NRF2 comprises the regulation of phase II enzymes which include the glutathione (GSH) thioredoxin, thioredoxin reductase 1, sulfiredoxin, and peroxiredoxin, which play an essential role in the reduction of oxidized protein thiols, as well as enzymes involved in NADPH generation, drug efflux, xenobiotic detoxification, and heme metabolism [66,67,68].

Years later after NRF2 description, KEAP1, an inhibitor of NRF2, was discovered [69]. KEAP1 induces NRF2 degradation under non-stressed conditions, whereas oxidative insults directly modify KEAP1 thiols groups, resulting in the inactivation of KEAP1 function, subsequent stabilization of NRF2, and induction of cytoprotective genes. Under normal conditions, NRF2 is constantly degraded via the ubiquitin-proteasome pathway in a KEAP1- dependent manner. Upon oxidative or electrophilic stress, the KEAP1 homodimer is inactivated leading to NRF2 stabilization and translocation to the nucleus. Nuclear NRF2 forms heterodimers with small Maf proteins and induces the expression of its target genes through binding to specific regions of the DNA named antioxidant response elements or electrophile response elements. NRF2 activates many cytoprotective genes [66]. Alternatively, KEAP1 is also regulated by the autophagy receptor, SQSTM1/p62, which acts both as a target and positive regulator of NRF2 [70,71]. SQSTM1/p62 can directly bind to KEAP1 [72]. On disruption of autophagy, there is the accumulation of SQSTM1/p62, which activates NRF2 by competing for the binding to KEAP1 [71]. Importantly, SQSTM1/p62 regulates NRF2 independently of the cellular redox state, which may connect NRF2 activity to the autophagy response pathway [61]. Overall, several mechanisms may lead to NRF2 activation and NRF2 is a key host factor in the host’s antioxidant response during an infection to limit over exacerbated tissue damage, however, this comes with a cost since its activation can favor pathogen persistence.

## 5. Interplay between Inflammation, NF-κB, and NRF2 Transcription Factors

Macrophages are myeloid innate immune cells that reside in many organs throughout the body and present distinguished tissue-specific cellular functions. They are specialized in the detection, phagocytosis, and destruction of harmful invading organisms [73]. Inflammation is triggered when host cells recognize conserved structures on pathogens, called microbial-associated molecular patterns (MAMPs), or endogenous stress signals such as ROS, called danger-associated molecular patterns through pattern recognition receptors (PRRs). These receptors are expressed by myeloid cells, such as monocytes, macrophages, neutrophils, and dendritic cells [74] as well as by several non-immune cells including epithelial cells and fibroblasts [75,76]. Classes of PRRs include membrane-bound Toll-like receptors (TLRs), C-type lectin receptors, cytosolic RIG-I-like receptors (RLRs), and Nod-like receptors (NLRs). Their activation leads to the induction of pro-inflammatory pathways [77]. In the context of *Leishmania,* most studies have focused on the role of TLRs.

Macrophages activated by intracellular pathogens have been classically designated as pro-inflammatory M1 macrophages. M1 macrophages are found in an inflammatory scenario that is controlled by the signaling of TLR and IFN pathways, which guide acute inflammatory responses with the release of pro-inflammatory cytokines such as IL-1β, IL-6, IL-12, IL-18, IL-23, and TNF-α. This pro-inflammatory signaling cascade induces Th1 response activation and facilitates complement-mediated phagocytosis [78,79]. On the contrary, M2 macrophages are responsible for the induction of the anti-inflammatory response with the production of cytokines including IL-10 and IL-13, or chemokines such as the C-C motif ligand 22. M2 macrophages have been described to particularly participate during parasitic, helminthic, and fungal infections [78].

The binding of MAMPs to TLRs initiates signaling cascades that induce the nuclear translocation of nuclear factor-kappa B (NF-κB) transcription factors leading to transcription of IL-6 and TNF-α cytokines, and type I IFN (IFN-I) [80]. The NF-κB is a family of transcription factors that consists of five members: p50, p52, p65 (RelA), c-Rel, and RelB. Dimerization of the NF-κB family is necessary for their DNA-binding properties. In unstimulated cells, NF-κB dimers are mainly cytoplasmic due to the binding of a set of inhibitory proteins known as the inhibitor of the NF-κB (IκB or IKK) family [81]. In contrast, in stimulatory conditions, such as infection, exposure to pro-inflammatory cytokines and oxidative species [82,83] will lead to the first step of NF-κB activation requiring post-translation modification of IκB inhibitors. Two defined mechanisms called the canonical pathway and, the alternative or non-canonical pathway, have been described for the induction of the NF-κB complex [84,85]. The pathways differ in the receptor inducing the signaling cascade, the NF-κB dimers, and the IKK components involved.

Contrarily to its activation, the mechanisms responsible for terminating the NF-κB pathway remain still poorly understood. Most of the studies have been centered on mechanisms involving IκB proteins and upstream signaling intermediates [86,87]. However, NF-κB signaling is also regulated by negative feedback mechanisms such as by A20 deubiquitinase (also known as TNFAIP3). A20 is essential for maintaining immune homeostasis and downregulating inflammation as confirmed by A20 deficient mice that prematurely die due to spontaneous multi-organ inflammation [88]. The inflammatory NF-κB signaling can also be controlled by NRF2 [89]. Indeed, numerous studies over the past years demonstrate the connection between the NRF2 and NF-κB pathways to regulate the transcription or function of downstream pro-inflammatory proteins [90,91,92].

Several mechanisms have been described on how NRF2 negatively regulates inflammation. The NRF2-Heme Oxygenase-1 (HO-1) axis not only helps the antioxidant response but also plays an anti-inflammatory role. NRF2 indirectly inhibits inflammation by HO-1 induction that inhibits NF-κB (i.e., RelA) phosphorylation at S276, a critical site for sustaining TNF-dependent NF-κB activation [93]. Interestingly, NF-κB activation is upregulated in *Nrf2*^−/−^ mice leading to acute inflammation [91]. Additionally, NRF2 regulates NF-κB activation by modulating the degradation of IκBα as described by using mouse embryonic fibroblasts deficient in NRF2 [91]. Accordingly, the NRF2 activator, sulforaphane (SFN), present in cruciferous vegetables like broccoli and cabbage, has been described as a negative regulator of inflammation by decreasing the expression of NF-κB-induced pro-inflammatory cytokines, such as IL-1β and TNF-α, and the inflammatory mediators cyclooxygenase 2 and iNOS [90]. Similarly, NQO1 activation downregulates the LPS-induced expression of proinflammatory cytokines [94]. The NRF2 activator and target protein p62/SQSTM1 has also been described as a negative modulator of the inflammatory pathway [95]. p62/SQSTM1 deficiency resulted in higher levels of the proinflammatory cytokine IL-1β. NRF2 and NF-κB pathways synergize to induce p62/SQSTM1 to counteract uncontrolled inflammation and prevent NLRP3-inflammasome activation [70]. The pro-inflammatory cytokine IL-17 is repressed by NRF2 in the case of autoimmune encephalomyelitis [96]. Furthermore, the induced levels of GSH by NRF2 have been reported to affect TNF-α levels [97].

NRF2 has also been connected with the activation of T cells. Activation of NRF2 in CD4+ T cells has been associated with decreased expression of activation markers such as CD25 and CD69 as well as reduced activation of NF-κB [98]. The maintenance of ROS and antioxidant protein balance within the cell is critical for keeping the integrity of T-cell mediated immunity, therefore NRF2 plays a crucial role in limiting T cell activation. Accordingly, NRF2 activation reduces IFN-γ production and raises IL-4 and IL-13 cytokines in CD4+ T cells, and skews them into a Th2 differentiation promoting the anti-inflammatory response [99,100]. Thus, NRF2 aside from having a clear antioxidant role could represent a key player in attenuating inflammation to limit pro-inflammatory responses and tissue damage in *Leishmania* infection.

## 6. *Leishmania* Parasites and Inflammation

Macrophages are crucial for disease progression and the favorable/unfavorable outcome depends on the interplay between the infecting *Leishmania* spp. and the type and magnitude of the host immune response [49,101]. During *Leishmania* infection, both pro-inflammatory M1 and anti-inflammatory M2 macrophages are induced. However, macrophage polarization phenotypes are not mutually exclusive which complicates the immune scenario during leishmaniasis [49]. Additionally, *Leishmania* parasites have developed strategies to subvert the pro-inflammatory response to promote parasite survival [101,102].

The immune response led by TLRs plays a crucial role in determining the response of the host during *Leishmania* infection [103]. Parasite surface molecules are recognized by TLRs. For example, LPGs are recognized by TLR-2 resulting in the activation of NF-κB [104,105]. Similarly, the dsRNA of LRV1 within *L. guyanensis* (*Lgy*) parasites induces the cascade of pro-inflammatory cytokines via TLR-3 signaling [18]. Bone marrow-derived macrophages (BMDMs) infected with *Lgy* parasites bearing the LRV1 endosymbiont demonstrate that LRV1 is recognized by TLR-3 promoting hyper-inflammation and IFN-I production resulting in tissue destruction and parasite persistence as observed in MCL patients [18]. These findings are supported by in vivo infection of mice deficient in TLR-3 (*Tlr3*^−/−^), which present a significant decrease in footpad swelling at the peak of infection in comparison to the control C57Bl/6 WT mice (Figure 2) [18]. Moreover, LRV1 presence is associated with IL-17 secretion, which contributes to LRV1-mediated disease severity (Figure 2). The IL-17 cytokine is produced by Th17 in hyperinflammatory situations, such as LCL. Interestingly, in vivo infection of IL-17 deficient (*Il17*^−/−^) mice with *Lgy*LRV1+ showed decreased LRV1-mediated pathology [17]. Furthermore, LRV1-induced TLR-3 activation has been associated with parasite persistence by promoting macrophage survival through AKT signaling partially relying on the microRNA miR-155. miR-155 has been described to be the only microRNA upregulated by the presence of LRV1. Infection of miR155 deficient (*Mir155*^−/−^) mice with *Lgy*LRV1+ results in decreased disease pathology, which indicates that LRV1 uses miR1-55 as another survival strategy [106]. The relationship between LRV1 and IFN-I (ie IFN-alpha and IFN-beta) production has been further investigated by Rossi et al., who demonstrate that the IFN-I response is responsible for worsening the outcome of leishmaniasis. Injection of recombinant IFN-I to mice infected with *Lgy*LRV1- or coinfection with lymphocytic choriomeningitis virus (LCMV) or Toscana virus (TOSV) resulted in higher lesion size and parasite load as well as downregulation of the IFN-γ receptor (IFN-γR), which is responsible for mediating the antileishmanial response induced by IFN-γ [7,107]. Similarly, *L. braziliensis* parasites bearing the LRV1 endosymbiont have been reported to promote aggressive pathogenesis in the context of HIV coinfection. The immunosuppressed immune status of HIV patients represents a favorable environment for disease exacerbation exerted by LRV1 [108]. The exact mechanism as to how LRV1 and HIV may synergize to worsen leishmaniasis outcomes has not, as yet, been described.

LRV1 signaling has also been studied in the context of other PRRs, such as the cytoplasmic dsRNA sensor RLRs and the inflammasome-independent NLRs [109]. RLRs are cytosolic receptors and promote the IFN-I response upon recognition of viral RNA [110]. No role for RLR-signaling in response to LRV1 was detected, which indicated that LRV1 was not likely to exit the phagolysosome to engage cytoplasmic receptors [109]. In contrast, LRV1 has been reported to induce the expression of inflammasome-independent NLRs, such as NLRC2 and NLRC5, which are known to induce anti-viral responses. NLRC2 has been described to direct monocytes to the infection site, which correlated to the phenotype observed in mice deficient in NLRC2 (*Nlrc2*^−/−^) infected with *Lgy*LRV1+ presenting decreased lesion size and parasite load [109]. NLRC5, on the other hand, regulates MHC-I expression. MHC-I peptide complexes bind to CD8 cytotoxic (CD8+) T cells. The role of CD8+ T cells during *Leishmania* infection has been controversial since discrepancies among infection with different strains have been debated [111]. In the case of *L. braziliensis* LCL, CD8+ T cells are described to mediate tissue injury [112]. However, NLRC5 deficiency in mice had no consequence on *Lgy*LRV1+ pathogenicity, which indicated that CD8+ T cells may not play a role in LRV1-mediated pathology in *L. guyanensis* infection. Additionally, no inflammasome activation was detected in vivo and in vitro in *Lgy*LRV1+ infection [109]. These findings confirmed that LRV1 induces the pro-inflammatory signaling pathway via TLR-3 in *L. guyanensis* LCL. Recently, it has been described that LRV1 subverts the NLRP3 inflammasome activation via TLR-3-induced autophagy in *L. guyanensis* and L. *braziliensis* infections [113].

Several mechanisms used by LRV1 to promote aggressive MCL pathology have been identified. However, the way how this viral endosymbiont is exposed to the host is still under debate. Recent studies have demonstrated that exosomes are secreted by *Leishmania* within the sand fly gut, and are transmitted along with the parasites during the sand fly blood meal. Additionally, LRV1 viral particles have been detected in *Leishmania* exosomes, indicating that LRV1, as many other viruses infecting higher eukaryotes [114] could take advantage of the exosomal pathway to be externally released and become better suited to interact with the macrophage dsRNA sensing machinery [115,116]. This more direct interaction with the host could be used by LRV1 to promote its propagation and infectivity towards its hosts, ultimately, resulting in severe leishmaniasis outcomes. Taken together, infection with *Leishmania* parasites can lead to the simultaneous activation of multiple PRRs leading to the induction of pro-inflammatory pathways to restrain infection, however, often leading to increased pathology.

## 7. NRF2 Role in *Leishmania* Infection

The antioxidant and anti-inflammatory pathways induced by NRF2 have been associated with resistance to infection with multiple pathogens including viruses, bacteria, and protozoan microorganisms such as *Entamoeba histolytica*, *Plasmodium* spp., *Toxoplasma gondii*, *Cryptosporidium parvum,* and *Leishmania* spp. [117,118,119,120]. The role of the NRF2 pathway in *Leishmania* infection has not been widely explored. Infection with *L. amazonensis* has been reported to induce NRF2 activation through dsRNA-dependent protein kinase (PKR). This kinase activates Phosphoinositide-3-kinase (PI3K)/AKT signaling triggering NRF2 release from KEAP1 as well inducing NRF2 activator and target protein p62/SQSTM1. Activation of NRF2 in *L. amazonensis* decreases oxidative stress levels and favors parasite survival as observed by infecting NRF2-knockdown macrophages [121]. Our own findings revealed that the KEAP1-NRF2 pathway is activated in different *Leishmania* spp. suggesting that the activation of the antioxidant response is conserved among species and the difference in oxidative levels is not crucial to mount the antioxidant response in *Leishmania* infection (Figure 2). Consequently, the induced antioxidant response favors parasite survival. At late phases of macrophage infection in *L. guyanensis* infection, *Nrf2*^−/−^ cells presented reduced parasite burden independently of LRV1. These data correlated with the increased expression of the antioxidant genes *Hmox1* and *Nqo1* regulated by NRF2 in *L. guyanensis* infection [122].

In *Lgy* infection, the ROS produced by NOX2 initiates the signaling responsible for NRF2 activation (Figure 2) [122]. Additionally, NOX2 regulation of NRF2 pathway is preserved among different *Leishmania* spp., confirming that *Leishmania* parasites have evolutionarily managed to exploit the NRF2 signaling pathway in the host cell in a similar way for their own benefit. Contrary to *L. amazonensis* infection, the PI3K/AKT axis does not participate in NRF2 pathway activation in *Lgy* infection as tested by pharmacological inhibition of either PI3K or AKT. Furthermore, PKR signaling is promoted by IFN-I and poly I:C, which increases expression and nuclear translocation of NRF2 as described in *L. amazonensis*. Poly I:C mimics LRV1 infection by inducing TLR-3 signaling. LRV1 has been described to be present in *L. guyanensis* and *L. braziliensis* but not in *L. amazonensis* [123]. Therefore, *L. amazonensis* is able to promote PKR independently of the presence of dsRNA, which could be linked to the ROS production by NOX2 [124].

Additionally, the flavonoid quercetin, which has antioxidant properties, also acts as an anti-leishmanial drug by reducing the parasite burden within macrophages. The NRF2/HO-1 pathway induced by quercetin results in upregulation of the ferritin complex, which controls the bioavailability of labile iron pool, impeding the uptake of this metal by *L. braziliensis*. Depletion of available iron decreases parasite replication and survival within macrophages [125]. 

Thus, NRF2 presents a dual role favoring both the parasite and the host by the induction of the antioxidant and anti-inflammatory response. Parasite contact with the macrophage seems sufficient to reprogram macrophage metabolism in response to *Leishmania* which induces the SRC-family kinase (SFK) signaling cascade triggering the activation of the NRF2 pathway (Figure 2) [122]. To sum up, the tripartite mutualism of *Leishmania*, LRV1, and macrophages favors parasite survival and therefore induces disease exacerbation. *Leishmania* parasites take advantage of the induced host detoxification machinery via the NRF2 pathway. In addition, the NRF2 pathway confers protection to LRV1/*Leishmania* by limiting the proinflammatory response. These results put into evidence the challenges in the design of specific drugs against *Leishmania* parasites and the important role of oncogenic kinases in leishmaniasis.

## 8. Perspectives

*Leishmania* parasites modulate host cell metabolism just 15 min post initial exposure [126]. The nature of the contact between *Leishmania* and the host cell surface molecules is not known. It could be relevant to define whether it is the flagellum that is responsible for the initial contact and to initiate the NRF2 pathway. *Leishmania* promastigotes possibly enter macrophages in a polarized manner through their flagellar tip and are then internalized into the host lysosomal compartments [126]. Interestingly, the initial parasite contact with the host cell is sufficient to mount the antioxidant response prior to phagocytosis [122].

In every *Leishmania* infection, the NRF2 pathway is induced by NOX2 signaling. Despite the large knowledge of NRF2 and its relationship with ROS, this is the first report of the link between NRF2 and the superoxide-producing NOX2 [122]. It also indicates that the activation of the antioxidant response is a general mechanism conserved among species.

Oxidative stimuli have been described to promote NRF2 via an SFK/PKCδ signaling circuit [127]. Consistently, in *Lgy* infection of macrophages, the expression of NRF2 depends on the NOX2/SFK/PKCδ axis. The NRF2 pathway is similarly activated by both *Lgy*LRV1+ and *Lgy*LRV1- parasites in macrophages and therefore the antioxidant response is promoted independently of the presence of LRV1 [122]. Furthermore, no TLRs are likely involved in NRF2 activation as shown by using mice deficient in the TLRs MyD88 and TRIF adaptors proteins. These findings reveal that *Leishmania* parasites could modulate the activation of the NRF2 pathway in macrophages by contact with a non-TLR pattern-recognition receptor such as DECTIN-1, which could mediate SFK signaling [128,129]. DECTIN-1 activation has already been described in the context *of L. amazonensis* [130]. Further studies are required to confirm the role of DECTIN-1 in this activation pathway.

Contrary to *Lgy*LRV1- parasites, which only activate NRF2, *Lgy*LRV1+ parasites induce both NRF2 and TLR-3-dependent inflammatory cytokines. This crosstalk between NRF2 and NF-κB pathways prevents hyper-inflammation due to high oxidative levels within the cell [89]. These data revealed that *Lgy*LRV1+ parasites not only take advantage of the antioxidant response from activation of the NRF2 pathway but also from the NRF2-dependent anti-inflammatory response [131]. In this regard, other *Leishmania* spp. impair NF-κB signaling and dampen host immune response. For example, *L. mexicana* promastigotes favor the cleavage of p65 NF-κB subunit generating a smaller p35 protein [132] and *L. amazonensis* activates the NF-κB p50/p50 repressor complex [133]. Similarly, mice deficient in NRF2 have significantly reduced disease pathology when infected with *Lgy*LRV1+, but not when infected with *Lgy*LRV1-. This validates that NRF2 participates in the control of the host pro-inflammatory response provoked by *Lgy*LRV1+ parasites. *Nrf2*^−/−^ mice have been shown to present higher levels of proinflammatory cytokines [134]. In this context, pro-inflammatory cytokine TNF-α levels are higher in *Nrf2*^−/−^ cells infected with *Lgy*LRV1+ parasites, and TNF-α levels are highly increased in MCL. TNF-α constitutes a clear risk factor for disease development and immunotherapies directed to its production, such as TNF-α blockers [135], have been proposed as a treatment against leishmaniasis [136]. Thus, TNF- α levels are crucial for the outcome of leishmaniasis and could be of interest to investigate the impact of anti-leishmania drugs on this pro-inflammatory cytokine.

The NRF2 pathway could confer tissue damage control and diseases tolerance to systemic infections [118]. Wounds produce large amounts of ROS to combat invading pathogens, as ocurred in *Leishmania* infection, resulting in immune cells attraction. NRF2 has been shown to become activated in tissue damage favoring wound repair [137]. *Nrf2*^−/−^ mice do not have an obvious skin phenotype under *Lgy*LRV1+ infection but NRF2 could play a role in chronic leishmaniasis as assessed in a metastatic model of leishmaniasis. The deficiency of the NRF2 protein in *Ifng*^−/−^ mice, and *Nrf2xIfng* double knock-out mice (dKO), promotes disease exacerbation and hyper-inflammation in *Lgy*LRV1+ infected mice. Additionally, the tails of these dKO mice show augmented cartilage destruction and increased cellular infiltration at the footpads, which could be induced by higher expression of matrix metalloproteases (MMPs) [122]. Indeed, studies with *Nrf2*^−/−^ mice with spinal cord injury display higher MMP9 activity [138]. Thus, IFN-γ is essential for controlling *Lgy*LRV1+ infection and the NRF2-mediated anti-inflammatory response and tissue damage control. Additionally, the hyperinflammatory response observed in dKO could be linked to the IL-17 cytokine, which has been described to be negatively regulated by NRF2 in autoimmune encephalomyelitis [96]. IL-17 secretion favors the dissemination of *Lgy*LRV1+ parasites over *Lgy*LRV1- and inversely correlates with IFN-γ inducing disease exacerbation [106]. The link between Il-17 and NRF2 would require further attention.

Overall, NRF2 presents a dual role in *Leishmania* infection favoring both the parasite and the host by the induction of the antioxidant and anti-inflammatory response. Initial contact of the parasite and the host is sufficient to reprogram macrophage metabolism in response to *Leishmania* which induces the SFK signaling cascade resulting in the NRF2 pathway activation. In this respect, developing targeted drugs towards inhibition of the SFK family could offer potential therapeutic candidates to treat and prevent leishmaniasis. These dangerous liaisons, the tripartite mutualism of *Leishmania*, LRV1, and macrophages favor parasite survival and therefore induce disease exacerbation. *Leishmania* parasites benefit from the increased survival of macrophages via the LRV1 induced AKT signaling pathway. Furthermore, *Leishmania* parasites take advantage of the induced host detoxification machinery via the NRF2 pathway and the NRF2 pathway confers protection to LRV1/*Leishmania* by limiting the proinflammatory response. Thus, this association benefits the parasite and the infected cell but not the host.

## Figures and Tables

**Figure 1 pathogens-11-00409-f001:**
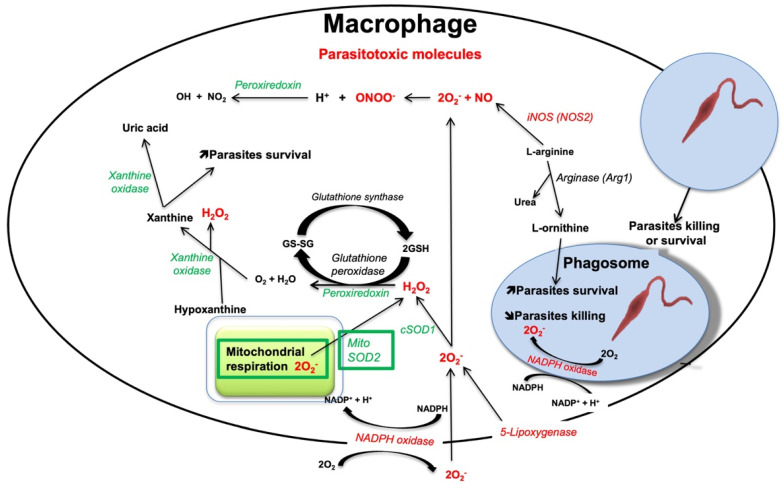
Representation of the different pathways and enzymes leading to the production of parasitotoxic molecules and enzymes (red) and of detoxification enzymes (green) in macrophages infected by *Leishmania* parasites.

**Figure 2 pathogens-11-00409-f002:**
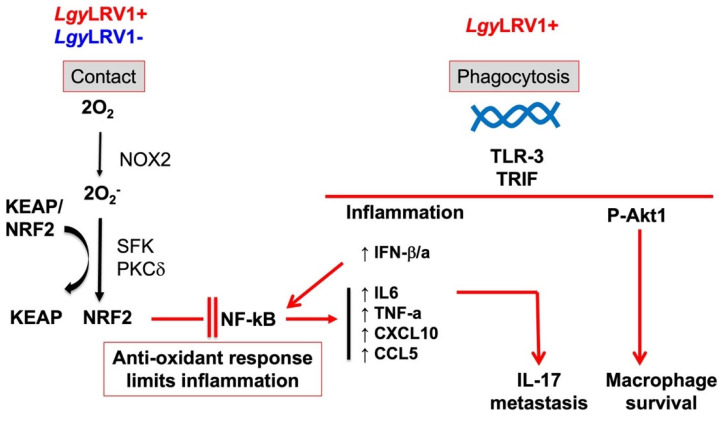
The oxidative stress and the inflammatory response in *Leishmania* infection. Independently of the presence of LRV1, the NRF2 pathway is activated upon contact between the parasite and the macrophage producing oxygen species generated by NOX2 permitting the release of NRF2 from its negative regulator KEAP1 and phosphorylation via SFK and PKC. This anti-oxidant response limits the NF-kB inflammatory and the production of inflammatory chemokines and cytokines. In the presence of LRV1 in *Leishmania* parasites, survival of infected macrophages is increased and production of Type-I interferon and inflammatory chemokines and cytokines are induced leading to accelerated dissemination of the infection via IL-17.

## Data Availability

Not applicable.

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
