# Peer review of "The Dangerous Liaisons in the Oxidative Stress Response to Leishmania Infection"

_pathogens, 2022, doi:10.3390/pathogens11040409_

Round 1
Reviewer 1 Report
In this review, the authors describe the complex interactions between Leishmania, the presence of a viral endosymbiont, and the host immune response. The authors clearly and thoroughly describe the genes involved in the oxidative stress response to Leishmania infection and the complex response of the immune response. Overall, I found the review to be very informative and interesting. Some small grammatical issues I noted in my review which I have listed below.
Line 42- no need to write cytokine
Line 46- delete increase
Line 50- Spell out MCL before first use
Multiple occasions- do not start a sentence with an abbreviation
Line 170- maybe rephrase to state that Leishmania can antagonize host immune response In lieu of more general "bail themselves out"
Line 179- no "an" necessary
Line 199- Nrf2 instead of NRF2
Line 236- Pathogen recognition receptors are expressed on a broad range of cells including epithelial cells and fibroblasts not just on myeloid immune cells
Consider using MAMP- microbial-associated molecular patterns rather than PAMP, a more accurate description that is being used in literature now.
Also, I have seen many studies shy away from the Th1/Th2 dichotomy since we now recognize more complexity among CD4 phenotypes.
Author Response
Please attachment

Reviewer 2 Report
Manuscript
The dangerous liaisons in the oxidative stress response to Leishmania infection
Marta Reverte, Tiia Snäkä and Nicolas Fasel
In their manuscript “The dangerous liaisons in the oxidative stress response to Leishmania infection”, Marta Reverte et al. provide a review on the role and importance of the phagocytic cell response to leishmania infection. The authors first present the leishmania and the oxidative response of phagocytic cells in relation to leishmania (chapter 2 and 3). This is followed by 4 long chapters (4-7) presenting the cellular response, which is not related to leishmaniasis and is more like an immunology review. Finally, comes 2 chapters more related to leishmania, with a chapter on the role of the LRV1 endosymbiont which is interesting. However, the information presented is not clearly synthesised, making it difficult to read. This lack of synthesis is largely due to the lack of data in the literature and the presence of many debates. Finally, the last chapter "perspectives" seems to try to bring together and synthesise the different information, but it remains very complex.
The review is seriously lacking illustrations that would allow the reader to follow more easily and also to have a more synthetic vision. Finally, this review is of limited interest and is too focused on the phagocytic cell with a limited link to leishmania.
Reviewer 3 Report
This study revises the relationship between the leishmanial parasite and the host macrophages, the role of oxidative stress response to the parasite and how anti-oxidant responses exhibit a dual role in Leishmania infection favoring both the parasite and the host.
The topic is of great interest to better understand the pathogenesis of Leishmania infection. However, the manuscript structure requires major revisions to increase clarity.
In details:
- Figure 1 is linked to section 2 - The oxidative stress response. However, figure 1 describes the role of Leishmania RNA virus (LRV) in the oxidative burst and how IFN-beta affects gene expression; neither LRV nor IFN-beta are described in the section 2 of the manuscript.
- In figure 1, no distinction is done between molecules that are up or downregulated by IFN-beta, which renders unclear the role of this cytokine. Furthermore, IFN-beta is mentioned in Figure 1, but not in the text of the review, and a generic IFN-I response is presented; description of different IFN type I molecules should be provided.
- A paragraph at the end of each section summarizing the major findings and how they connect to the topic of the review would increase clarity.
- The Section 10. “Perspectives” should be shortened and focused on challenges and perspectives
- Language editing is
Minor comments:
Line 45: “interleukin (IL) 10 (IL-10)” should read “interleukin (IL)-10”;
Lines 55-57: “MCL patients present higher percentage of TNF-α and lower levels of IL-10, which results in the hyperinflammatory response characteristic of MCL lesions” ; not clear what “higher percentage” stands for. In addition the second term of comparison is missing.
Line 62 and line 84: Dot is missing at the end of the sentences;
Line 72: CDL should read DL;
line 176: “the nuclear factor-erythroid factor 2-related factor 2 (NRF2) transcription factor” should read “nuclear factor-erythroid 2 related factor 2 (NRF2)”;
lines 248-249: “or chemokines such as CCL and CCL22”; specify the first chemokine as there is no number after CCL;
lines 578-9: “In this respect, developing targeted drugs towards inhibition of SFK family that could offer potential therapeutic candidates to treat and prevent leishmaniasis” should read “In this respect, developing targeted drugs towards inhibition of SFK family could offer potential therapeutic candidates to treat and prevent leishmaniasis”.
Round 2
Reviewer 2 Report
The authors have done a very good job of synthesizing and reorganizing the manuscript. They also focused more on leishmania. The manuscript is now more easily readable. However, it is a shame that there are not more illustrations.
Author Response
We understnd the commeents on adding a figure but we did not find a additional good figure to illustrate our text.
Reviewer 3 Report
Minor comments
- lines 207-210: Figure 1 should be cited also in this paragraph.
- line 283-4: “Interestingly, NF-κB activation is upregulated in Nrf2 -/- mice leading to in acute inflammation “; “in” should be removed
- Line 321: “Lgy” should read “L.guyanensis (Lgy)”
- Line 338: “The relationship between LRV1 and IFN-I production has been further investigated” should read “The relationship between LRV1 and IFN-I (ie IFN-alpha and IFN-beta) production has been further investigated”
- Lines 347-8: “The exact mechanism how LRV1 and HIV may synergize to worsen leishmaniasis outcome has not been described yet” should read “The exact mechanism by which LRV1 and HIV may synergize to worsen leishmaniasis outcome has not been described yet.”
- Line 508: “resulting in NRF2 pathway” should read “resulting in activation of the NRF2 pathway”
Author Response
There were only minor corrections. Please see the "highlighted" version with modifications in red and downloaded as a PDF